# Cyclophilin D knockout significantly prevents HCC development in a streptozotocin-induced mouse model of diabetes-linked NASH

**Winston T. Stauffer**[1], **Michael Bobardt**[1], **Daren R. Ure**[2], **Robert T. Foster**[2], **Philippe Gallay**[1] *

**1** Department of Immunology & Microbiology, Scripps Research, La Jolla, California, United States of America, **2** Hepion Pharmaceuticals, Edison, New Jersey, United States of America

* gallay@scripps.edu

**Data Availability Statement:** All relevant data are within the manuscript.

**Funding:** PG 5R01AI143931-04 Funding for this study was supplied by a grant from the National

## Abstract

A family of Peptidyl-prolyl isomerases (PPIases), called Cyclophilins, localize to numerous intracellular and extracellular locations where they contribute to a variety of essential functions. We previously reported that non-immunosuppressive pan-cyclophilin inhibitor drugs like reconfilstat (CRV431) or NV556 decreased multiple aspects of non-alcoholic fatty liver disease (NAFLD) in mice under two different non-alcoholic steatohepatitis (NASH) mouse models. Both CRV431 and NV556 inhibit several cyclophilin isoforms, among which cyclophilin D (CypD) has not been previously investigated in this context. It is unknown whether it is necessary to simultaneously inhibit multiple cyclophilin family members to achieve therapeutic benefits or if loss-of-function of one is sufficient. Furthermore, narrowing down the isoform most responsible for a particular aspect of NAFLD/NASH, such as hepatocellular carcinoma (HCC), would allow for more precise future therapies. Features of human diabetes-linked NAFLD/NASH can be reliably replicated in mice by administering a single high dose of streptozotocin to disrupt pancreatic beta cells, in conjunction with a high sugar, high fat, high cholesterol western diet over the course of 30 weeks. Here we show that while both wild-type (WT) and Ppif-/- CypD KO mice develop multipe severe NASH disease features under this model, the formation of HCC nodules was significantly blunted only in the CypD KO mice. Furthermore, of differentially expressed transcripts in a qPCR panel of select HCC-related genes, nearly all were downregulated in the CypD KO background. Cyclophilin inhibition is a promising and novel avenue of treatment for diet-induced NAFLD/NASH. This study highlights the impact of CypD loss-of-function on the development of HCC, one of the most severe disease outcomes.

## Introduction

Metabolic syndrome (MetS) is a multi-faceted disease, which combines the features of dyslipidemia, insulin resistance, and hypertension, is associated with poor diet, sedentary lifestyle, and genetic predisposition [1]. While effects of MetS are felt throughout the body, some of the most

Institutes of Health/National Institute of Allergy and Infectious Diseases NIH/NIAID https://www.niaid.nih.gov/ The funders had no role in study design, data collection and analysis, decision to publish, or preparation of the manuscript.

**Competing interests:** Daren R. Ure and Robert T. Foster are employees of Hepion Pharmaceuticals, the developers of Reconfilstat/CRV431.

acute effects are seen in the liver. The hepatic manifestation of MetS is non-alcoholic fatty-liver disease (NAFLD), which can encompass a range of symptoms, but is centered on steatosis, excessive lipid deposition in the liver. A more advanced form of NAFLD, non-alcoholic steato-hepatitis (NASH), combines steatosis with inflammation and may or may not also feature liver fibrosis. At this stage, the disease can be reversed with lifestyle changes and, potentially, with emerging drug treatments. Further progression however can lead to an increasingly hepatotoxic environment and eventually irreversible cirrhosis [2–6]. Notably, this stage of the disease can also become carcinogenic, leading to spreading nodules of hepatocellular carcinoma [7, 8].

While the exact pathogenesis of HCC remains unclear, it is likely related to the chronic inflammatory environment of late-stage NASH. Lipid-impaired hepatocytes are targeted by inflammatory immune cells, eventually causing DNA damage and the overactivation of DNA repair pathways. Errors in the repair process, along with other epigenetic alterations, eventually become carcinogenic, allowing liver cells to replicate out of control [8–10]. While HCC is also associated with viral infection, as in HBV or HCV-induced hepatitis [11–14], as well as alcohol [15] or drug-induced liver injury [16], HCC is increasingly known to be a fatal outcome of NAFLD/NASH [17]. HCC is the most common cause of death for those with liver cirrhosis and is overall the second leading cause of cancer deaths world-wide [18]. Diagnosis, typically made via biopsy or more recently via non-invasive imaging, often only occurs in late stages of the disease, decreasing patient survival and limiting treatment to surgical intervention, either liver resection or transplantation [2, 19]. Effective drug treatments are therefore needed to prevent HCC formation in the first place.

Previous studies have identified cyclophilin inhibition as a promising therapeutic avenue for the prevention of HCC in mice, both in the context of NAFLD/NASH [11, 12, 20] and viral hepatitis [14, 21, 22]. In separate reports, the sanglifehrin-derivative compound NV556 [11] and the cyclosporine A-analog CRV431 [20] were found to significantly limit multiple features of NAFLD/NASH in mouse models of the disease, including reducing HCC tumor burden. Both compounds are pan-cyclophilin inhibitors meaning they have been demonstrated to effectively abolish the activity of multiple cyclophilin family members [20, 23]. Cyclophilins are a family of peptidyl-prolyl isomerases (PPIases) that play important roles in various cellular functions [24, 25]. Cyclophilin A (PPIA, CypA) is a cytosolic enzyme which can be secreted and has been shown to play important roles in inflammation [26, 27] and the lifecycle of multiple viruses such as HIV, HCV, and HBV [27–30]. Cyclophilin B (PPIB, CypB) localizes to the endoplasmic reticulum where it serves as a chaperone, aiding in the folding of nascent proteins such as collagen [31–34]. Cyclophilin D (PPIF, CypD) is a component of the mitochondrial permeability transition pore (mPTP) where it can influence pore sensitivity and mitochondrial permeability [35–37]. In this capacity CypD has been shown to regulate cell death pathways [38–40], specifically those associated with oxidative stress [36, 41]. Because NV556 and CRV431 inhibit all cyclophilins, it is unknown whether inhibition of a specific family member is the most responsible for the previously observed beneficial effects in the context of NAFLD/NASH and the development of HCC.

In this study we isolated the effects of three cyclophilin family members by comparing a long-term diabetes-linked mouse model of NAFLD/NASH [42] across three mouse lines in which either CypA, CypB, or CypD had been globally deleted from all tissues from conception (KO). While CypA and CypB KO mice suffered high rates of mortality, sufficient CypD KO mice survived to be analyzed. While CypD KO mice developed multiple features of NAFLD/NASH, they were significantly protected from the development of HCC, both in terms of tumor size and number and overall resembled mice in an earlier stage of the disease compared to wild-type mice.

## Methods and methods

### Laboratory animal use

The data presented here involving the use of laboratory animals were generated in accordance with the Institutional Animal Care and Use Committee (IACUC) of Scripps Research and conforms to the rules provided by the National Research Council's Guide of the Care and Use of Laboratory Animals. This study was conducted under Animal Use Protocol 11-0015-5 "Cyclophilin Inhibitors and Hepatic Disease Models". Animals were monitored daily by Scripps Department of Animal Resources (DAR) staff and Dr. Stauffer who had received training on animal handling by Scripps DAR. For all listed experiments, mice displaying signs of severe discomfort were to be euthanized immediately upon recommendation of DAR veterinarians and staff. Specific clinical signs included weight loss, hunched body posture, lack of grooming, and dehydration. Mice are sometimes found dead of unknown causes despite not previously showing these signs. Throughout the course of the experiment 1 mouse was found dead out of a total of 60 mice. The remaining 59 animals were sacrificed at the end of the experiment. All animals sacrificed were euthanized by cervical dislocation while under full anesthesia via a nosecone with 2% isoflurane/O2. Animals were optimally kept with up to four cage-mates or five mice per cage. However, because all mice in the experiment were male, if a mouse were single-housed due to the death of a cage-mate(s), it was not possible to transfer it to another cage.

### CCl$_4$ model of liver fibrosis

Liver fibrosis was induced in mice using the hepatotoxic agent carbon tetrachloride (CCl$_4$) (cat# 270652, Sigma-Aldrich, St. Louis, MO, USA), as previously described [43]. Briefly, 4-week-old male C57BL/6J mice nourished with normal chow and water were given intraperitoneal (IP) injections of 0.2 µL/g CCl$_4$ twice weekly for 20 weeks. During each IP injection, mice were briefly anesthetized in an induction chamber using a 2% isofluorane/O$_2$ mixture to minimize animal movement and the risk of needle sticks. The CCl$_4$ was administered as an 8% CCl$_4$/corn oil solution. Therefore, for a typical 20 g mouse, 50 µL of the solution was administered with each IP injection. A fresh 8% CCl$_4$/corn oil solution was created each week. Control mice were given identical IP injections of pure corn oil. After 20 weeks, mice were anesthetized, sacrificed by cervical dislocation, and the livers weighed and removed. Livers were dissected into equal halves, one of which was flash frozen in liquid nitrogen and stored at -80˚C for later molecular analysis. The other half was suspended in zinc-buffered formalin fixative (cat# 5701ZF, Thermo Fisher Scientific, Waltham, MA, USA) for three days before it could be mounted in a paraffin block for histological analysis.

### STZ-WD model of NAFLD/NASH

Features of the human diseases NAFLD and NASH were reproduced in mice by giving a 200mg/kg bolus IP injection of streptozotocin (STZ) (cat# S0130, MilliporeSigma, Burlington, MA, USA) at 20mg/mL in 0.1M sodium citrate solution (pH 4.5) at 4 weeks of age. For example, a typical 15g mouse would receive 150L of the STZ solution. STZ injection destroys insulin-producing pancreatic beta-cells. STZ mice were then nourished with high-fat, high-sugar, and high-cholesterol western-diet chow (cat# TD.120528, Envigo Teklad, Madison, WI, USA) and sugar water, as previously described [42]. Briefly, 10-week-old male mice C57BL/6J mice received *ad libitum* western-diet chow containing 21.1% fat, 41% sucrose, and 1.25% cholesterol by weight, in lieu of normal chow. They simultaneously received *ad libitum* sugar-water containing 23.1 g/L fructose (cat# F0127, Sigma-Aldrich) and 18.9 g/L glucose (cat# G8270, Sigma-Aldrich), instead of normal water. Sugar water was prepared in advance, first as an

autoclaved 10X stock solution, and then as a 1X working solution which was again autoclaved in the water bottles provided by the Scripps Research vivarium. This diet continued for 30 weeks. After this period, mice were anesthetized, sacrificed by cervical dislocation, and the livers removed, weighed, and assessed from the presence of HCC (see below). Livers were then dissected into equal halves, one of which was flash frozen in liquid nitrogen and stored at -80˚C for later molecular analysis. The other half was suspended in zinc-buffered formalin fixative (cat# 5701ZF, Thermo Fisher) for three days before it could be mounted in a paraffin block for histological analysis.

### PCR array

PCR arrays were performed on cDNA transcribed from RNA isolated from mouse livers as described above, and generated using Qiagen RT2 First Strand Kit (cat# 330401, Qiagen, Germantown, MD, USA). RT2 Profiler PCR Arrays for Mouse Fibrosis (cat# PAMM-120Z, Qiagen, Germantown, MD, USA) were used according to the manufacturer's instructions.

### Histology

Liver tissue was fixed in zinc-buffered formalin as described above. Livers were then rinsed in 70% ethanol before an hour-long 70% ethanol bath, two hour-long 95% ethanol baths, two hour-long 100% ethanol baths, two hour-long xylene baths, and two four-hour-long liquid paraffin baths. Tissues were then placed in molds with more paraffin to create paraffin blocks suitable for sectioning on a Leica RM2125 microtome. Sections were typically 7mm thick and were incubated on glass histology slides overnight in a 60˚C oven.

### Picrosirius red staining

Prepared slides were deparaffinized in a series of three xylene washes, three 100% ethanol washes, two 70% ethanol washes, and two water washes. Slides were then stained for ten minutes in Weigert's Hematoxylin A (cat# 26044–05, Electron Microscopy Sciences, Hatfield, PA, USA) for nuclei, washed with water, stained for one hour with picrosirius red (cat# 26357–02, Electron Microscopy Sciences, Hatfield, PA, USA) for collagen fibrosis, and washed with acetic acid (cat# 10042–05, Electron Microscopy Sciences, Hatfield, PA, USA). Slides were then dehydrated in the ethanol and xylene washes in reverse order and covered with Permount (cat# SP15-500, FisherSci, Waltham, MA, USA) and a coverslip. Slides were imaged at 4X with a slide scanner. Fibrosis was quantified as a percent of total area using ImageJ software.

### Hematoxylin and eosin (H&E) staining

Prepared slides were deparaffinized as above. Slides were then stained for ten minutes in Gill's Hematoxylin (cat# 72511, Thermo Scientific) for nuclei, washed with water, dipped briefly in Eosin (cat# 7111, FisherSci) cytoplasmic stain, and washed in 1% acid alcohol (cat# 26072–01, Electron Microscopy Sciences). Slides were then dehydrated and prepared as above. Slides were imaged at 10X and 20X with a slide scanner.

### NAFLD/NASH scoring

H&E images were given a NAFLD Activity Score (NAS), including ballooning, inflammation, and steatosis. Our scoring system for NAS is as follows: ballooning: 0 (no changes), 1 (few ballooned cells), and 2 (many prominent balloon cells); for inflammation: 0 (no changes), 1 (minimal infiltration with no major inflammatory foci), 2 (mild with $<2$ inflammatory clusters), 3 (moderate with 2–4 foci of leukocytes), and 4 (severe infiltration with $>4$ inflammatory

clusters); for steatosis: 0 ($<$5% steatosis), 1 (5–33%), 2 (33–66%), and 3 ($>$66% steatosis). Scores for each category were added together and a score greater than 6 was considered indicative of NASH.

## HCC scoring

Immediately upon excision, livers were assessed for the presence of HCC nodules. The number and size of the nodules were recorded before livers were fixed or frozen. Nodule size was considered small if less than 0.5cm, medium if 0.5cm or larger but less than 1.0cm, and large if 1.0cm or larger. The largest nodule observed was 2.0cm across. Our scoring system for HCC tumor burden is as follows: 0 (no nodules present); 1 (fewer than five small nodules); 2 (five or more small nodules); 3 (one or two medium nodules and unlimited smaller nodules); 4 (three or more medium nodules and unlimited smaller nodules); 5 (one large nodule and unlimited smaller nodules); 6 (two large nodules and unlimited smaller nodules); 7 (three or more large nodules and unlimited smaller nodules).

## WT and KO mice

*Ppif* knockout male mice used in this study are commercially available from Jackson Laboratory. B6;129-Ppiftm1Jmol/J mice were generated so that one *Ppif* allele has had exons 1–3 globally deleted in all tissues and cell types. Heterozygous *Ppif* +/- mice were then cross-bred to generate homozygous *Ppif* -/- mice which have a complete absence of any CypD protein but still have all the other cyclophilin family members. Age-matched wild-type male C57BL/6J mice were bred and purchased from the Rodent Breeding Colony at Scripps Research. PCR primers used for genotyping are as follows.

*Ppif* Wild type–Fwd–5'–CTC TTC TGG GCA AGA ATT GC– 3'
*Ppif* Mutant–Fwd–5'–GGC TGC TAA AGC GCA TGC TCC– 3'
*Ppif*–Rev–5'–ATT GTG GTT GGT GAA GTC GCC– 3'

## Statistics

All error bars shown are ± standard error of the mean and statistical treatments were generated by multiple unpaired student's t-test, comparing WT and KO values across each experimental set independently.

# Results

## Non-diseased CypD KO mice are morphologically similar to WT

We first obtained a mouse line in which the CypD gene, *Ppif*, had been deleted in all tissues [44]. This line was then separated into three sets and paired with separate WT C57BL/6J controls. All three sets were maintained for 30 weeks. Set 1 was nourished on *ad libitum* normal water and chow and served as a non-diseased baseline. Set 2 also received normal water and chow but was subjected to a carbon tetrachloride ($CCl_4$) model of liver fibrosis [43]. Set 3 was tested in a diabetes-linked model of NAFLD/NASH [42].

Set 1 non-diseased CypD KO mice were viable and had no gross morphological differences from their WT counterparts. Body weights were not significantly different between KO and WT mice and liver weight and appearance were also normal. Set 1 CypD KO livers were free of steatosis, inflammation, and fibrosis and were overall similar in appearance to Set 1 WT livers, with a smooth, glossy surface and dark-brown to dark-red in color (Fig 1). This is consistent with many previous reports that CypD deficient mice are not obviously impaired despite CypD's role as a component of the mPTP and as a regulator of oxidative stress.

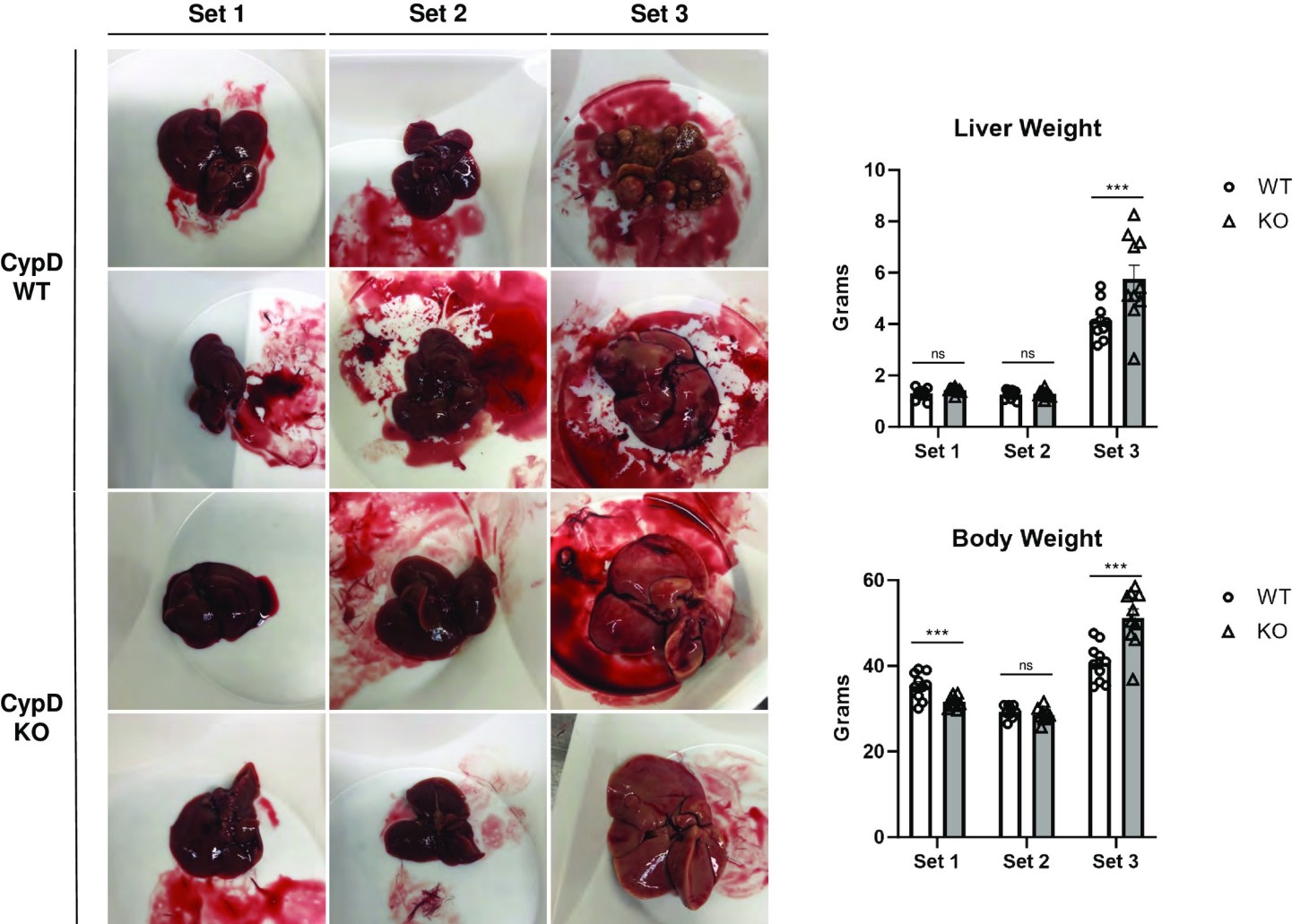

**Fig 1. CypD KO mice have similar liver morphology to WT mice except for the appearance of HCC tumors in the STZ-WD model.**

WT and CypD KO mice were separated into three sets. Set 1 mice were naïve, Set 2 received twice weekly CCl4 via intraperitoneal (IP) injection for thirty weeks, and Set 3 mice received a single STZ IP injection and were maintained on western diet (WD) for thirty weeks. Mice were sacrificed and then weighed, and the livers removed, weighed, and imaged. Set 1 livers were typically dark red with smooth surfaces. Set 2 livers were similar in size and color but had a slightly scaly surface. Many of the livers in Set 3 developed HCC tumors. Set 3 livers without tumors were larger and tan in color. Set 3 livers with tumors tended to be much smaller. Likewise, Set 3 mice tended to be larger overall, but mice with extensive HCC weighed less than mice without HCC. Representative livers for each group are shown here. ***$p \leq 0.001$ significance between a condition and WT control by unpaired students t-test.

## CypD KO mice develop similar liver fibrosis to WT in a CCl4-induced fibrosis model

Set 2 mice were given the same diet as Set 1 but received biweekly IP injections of 0.2μl/g CCl$_4$ to induce liver fibrosis. CCl$_4$ is a hepatotoxic agent which is useful in isolating the effects of an

intervening maneuver, in this case *Ppif* gene deletion, on liver fibrosis alone. Fibrosis is a common feature of advanced NASH and cirrhosis and pan-cyclophilin inhibition has previously been shown to reduce fibrosis staining in the same $CCl_4$ model. Set 2 WT and KO mice tolerated the biweekly IP injections well and nearly all mice from both groups survived to be analyzed at the end of the 30-week period. Upon extraction, Set 2 livers from both groups were similar in size to Set 1 livers, but had a slightly scaly surface and dark brown color (Fig 1). Sirius red staining of fixed livers showed branching interlobular fibrosis, but little to no steatosis or lobular inflammation, as expected in this model. WT and KO livers exhibited similar patterns of fibrosis, both in structure and in overall quantitation (Fig 2). Thus, CypD KO has no effect on liver fibrosis in this model, suggesting that the decrease in fibrosis previously observed with pan-cyclophilin inhibitor compounds NV556 and CRV431 came as a result of the inhibition of cyclophilin family members other than CypD.

WT and CypD KO Set 1, 2, and 3 mouse livers were fixed and stained with picro-sirius red to examine lobular fibrosis. Set 1 livers had minimal fibrosis except around blood vessels, as expected. Set 2 livers had significant branching fibrosis that did not differ between WT and CypD KO mice. Set3 mice also had significant fibrosis, which tended to be disrupted by the presence of macrovesicular steatosis and the presence of HCC tumors. Despite this, there was no significant difference in total liver fibrosis between WT and CypD KO Set 3 mice. The area of stained tissue relative to the total area was quantified with ImageJ software over several fields per sample. Representative images are shown here. ***$p \leq 0.001$ significance between a condition and WT control by unpaired students t-test.

## CypD KO mice develop drastically less HCC tumor burden compared to WT in a diabetes- and diet-linked model of NAFLD/NASH

Set 3 mice received a single bolus injection of streptozotocin (STZ) at 10 weeks of age to destroy pancreatic beta cells and induce insulin insufficiency. The mice were then fed an *ad libitum*

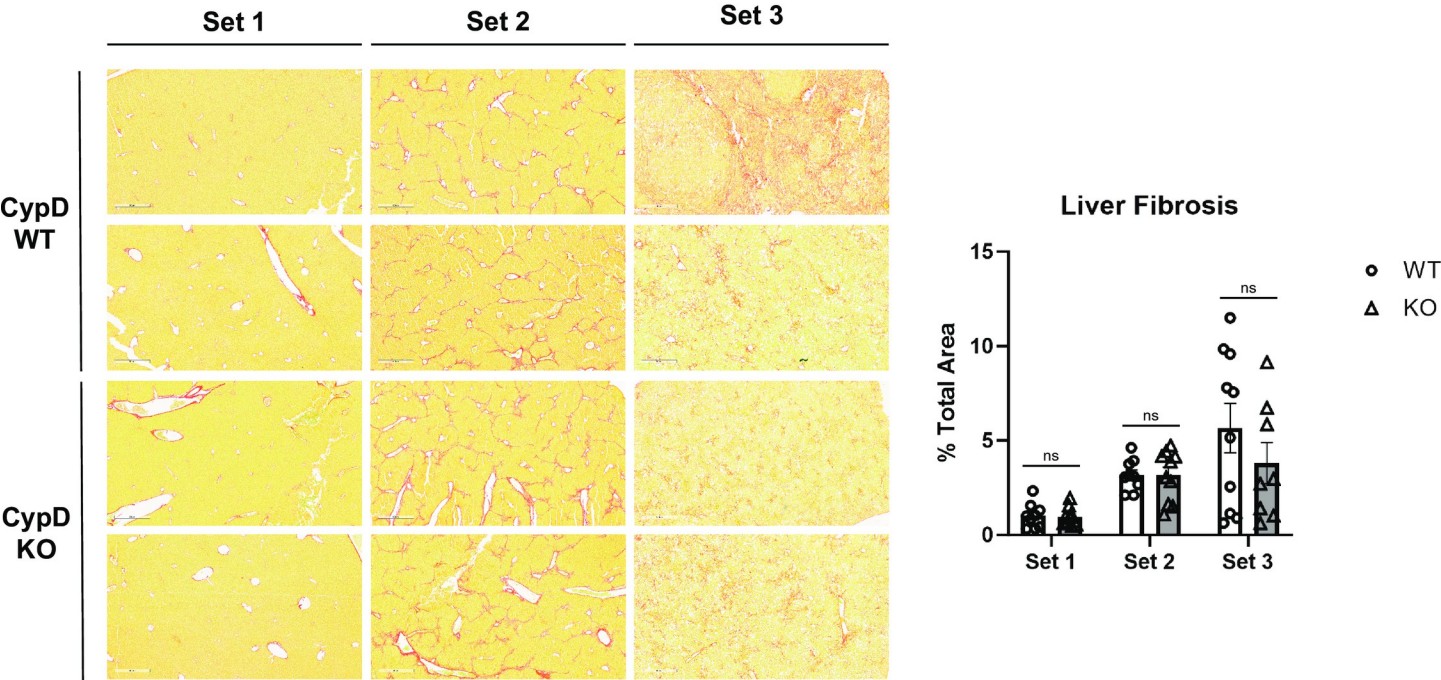

**Fig 2. CypD KO mice develop liver fibrosis similar to WT after administration of a CCl4 model or a STZ-WD model.**

western diet (WD) of high fat, high cholesterol chow and sugar solution in lieu of water for 30 weeks. This STZ-WD treatment program has been previously shown to reliably replicate multiple features of NAFLD/NASH seen in human patients. The STZ-WD model can also produce HCC if continued for a longer duration, such as our 30-week time frame. WT and CypD KO mice tolerated this model well and all Set 3 mice survived to the end of the experiment. Both WT and KO Set 3 mice gained weight relative to Set 1 mice over the course of the experiment, as expected from extended *ad libitum* western diet, but only KO mice were significantly different. Set 3 KO mice were also significantly heavier than their Set 3 WT counterparts. Set 3 liver weights more than quadrupled relative to Set 1, and again Set 3 KO livers were significantly heavier than WT (Fig 1). 70% of Set 3 WT livers exhibited some form of visible HCC and a majority were extensively covered in nodules. Visible nodules ranged in size from 0.1 to 2cm and appeared as pale, pea-shaped tissues on the exterior of the organ. Liver tissue not covered by nodules was a pale tan, typical of mice maintained on western diet. Strikingly, 80% of Set 3 KO livers were completely devoid of HCC nodules. Overall, Set 3 WT livers averaged 8.8 tumors per liver, while Set 3 KO livers averaged significantly fewer, only 1.6 per liver. Livers were also scored for the presence, quantity, and size of HCC tumors to quantify overall tumor burden for each group. Our HCC scoring criteria [11, 20] are summarized in Table 1.

Set 3 WT livers averaged a score of 3.2 while Set 3 KO livers averaged significantly less with a score of 0.8 (Fig 3). This indicates livers from mice without CypD are much less prone to the development of HCC even after a prolonged diabetes-linked NAFLD/NASH regimen. This is also in line with our previous studies showing that pan-cyclophilin inhibitors NV556 and CRV431 significantly reduced HCC tumor burden in NAFLD/NASH mouse models and suggests that specifically CypD inhibition was at least in part responsible for this effect.

Set 3 WT and CypD KO mouse livers were examined for the development of HCC tumors after 30 weeks of the STZ-WD model. HCC tumors appeared as pale, round nodules either on or embedded in the surface of the livers (white arrows). Tumors were assessed for size and quantity and scored according to the criteria in Table 1. Most WT set 3 mice developed significant tumor burden, while most CypD KO set 3 mice developed none at all. Representative images are shown here. ***p≤0.001 significance between a condition and WT control by unpaired students t-test.

## WT and CypD KO mice both develop similar characteristics of NAFLD/NASH

To determine whether Set 3 mice had indeed developed fatty liver disease as predicted, fixed livers were sectioned and stained with hematoxylin & eosin stain for analysis of the classical markers of NAFLD/NASH lobular inflammation, hepatocyte ballooning, and steatosis. Livers

**Table 1. HCC scoring criteria.**

| Score | Description |
|---|---|
| 0 | No nodules found |
| 1 | Fewer than five small (≤0.5cm diameter) nodules |
| 2 | Five or more small nodules |
| 3 | One or two medium (>0.5cm and <1.0cm) nodules and unlimited smaller nodules |
| 4 | Three or more medium nodules and unlimited smaller nodules |
| 5 | One large (≥1.0cm) nodule and unlimited smaller nodules |
| 6 | Two large nodules and unlimited smaller nodules |
| 7 | Three or more large nodules and unlimited smaller nodules |

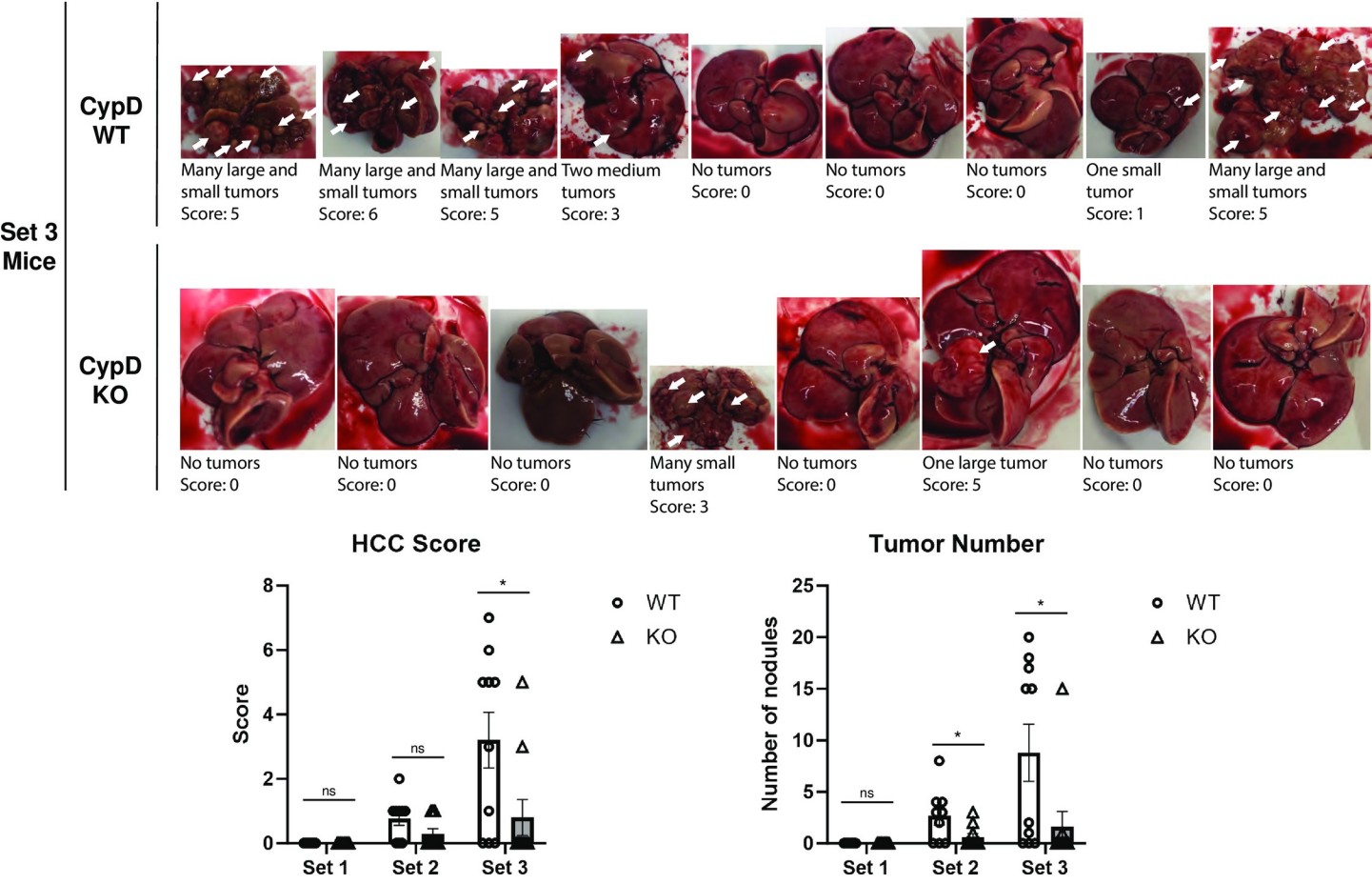

**Fig 3. CypD KO mice develop significantly less HCC tumor burden in the STZ-WD model.**

were each assigned a NAFLD Activity Score (NAS). A NAS of 6 or greater is considered to be consistent with a NASH diagnosis in human patients. Our NAS scoring criteria [11, 20] are summarized in Table 2.

In cases where HCC tumors were visible, only the non-cancerous tissue was assessed for NAS. As expected, Set 1 livers from both WT and KO mice had a NAS of near zero. However, Set 3 livers from both WT and KO mice exhibited a NAS of greater than 6, indicating the development of NASH. Both Set 3 WT and KO livers were indistinguishable from one another in all three NAS components (Fig 4). Thus, the model of diabetes-linked NAFLD/NASH was successful in producing mouse livers with the characteristics of NASH in both WT and KO mice. However, this means the relative lack of HCC development in Set 3 KO livers occurred in spite of the development of NASH comparable to Set 3 WT.

**Table 2. NAFLD activity score criteria.**

| Score | Lobular Inflammation | Hepatocyte Ballooning | Steatosis |
|---|---|---|---|
| 0 | No foci | No ballooned cells | <5% total area |
| 1 | 1 focus | Few ballooned cells | 5–33% total area |
| 2 | 2–4 foci | Many ballooned cells | 33–66% total area |
| 3 | >4 foci | | >66% total area |

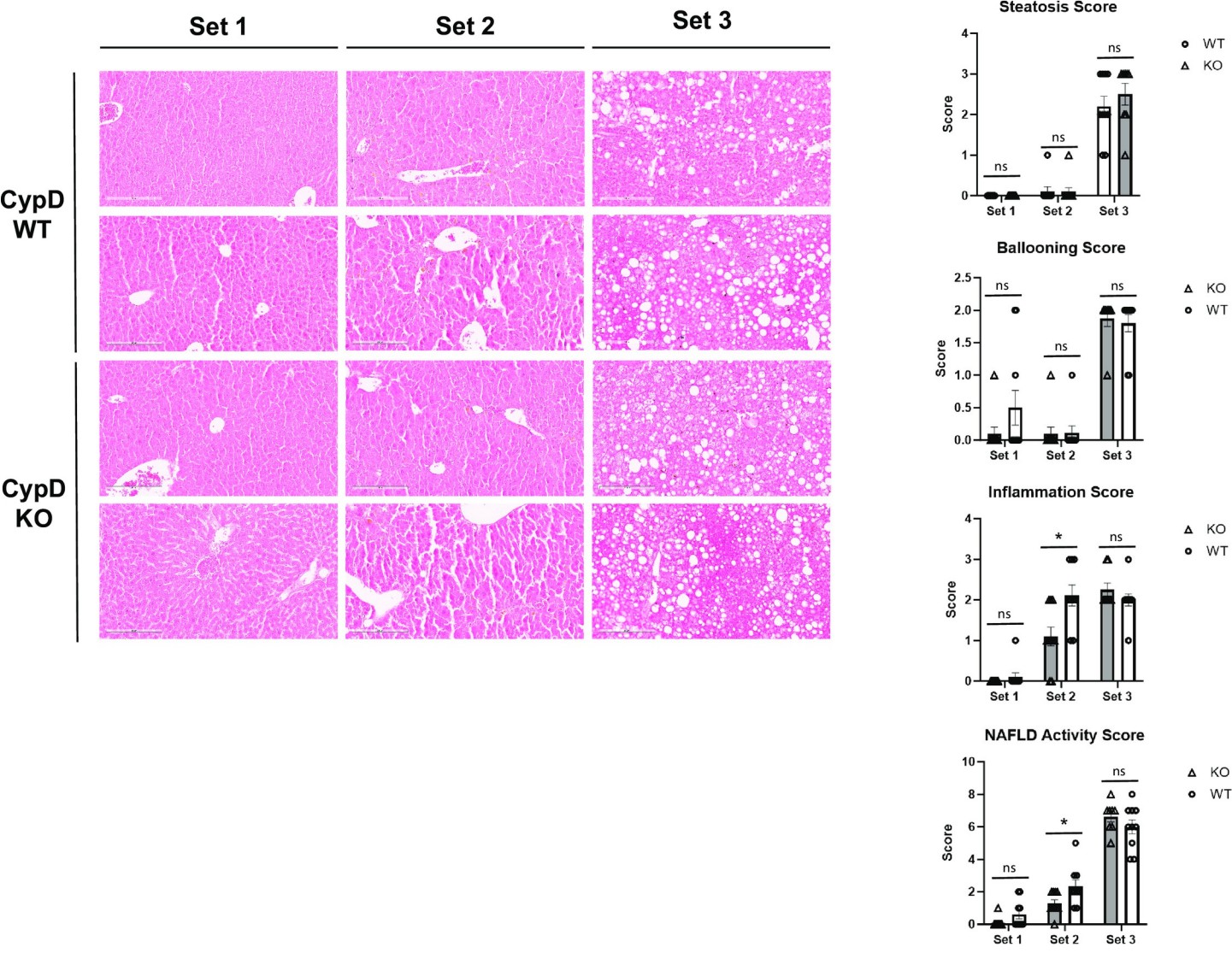

**Fig 4. CypD KO mice develop features of NAFLD/NASH similar to WT after administration of the STZ-WD model.**

WT and CypD KO Set 1, 2, and 3 mouse livers were fixed and stained with H&E to examine three key features of NAFLD/NASH: steatosis, hepatocyte ballooning, and inflammation. Set 1 and 2 livers developed minimal steatosis and ballooning, as expected. WT Set 2 livers developed significant inflammatory centers, which were significantly reduced in CypD KO livers. As expected, Set3 mice displayed the most significant NAFLD/NASH symptoms, with prominent steatosis, hepatocyte ballooning, and inflammation. However, there was no significant difference between WT and CypD KO. NAFLD Activity Scores were assigned based on the criteria in Table 2. Representative images are shown here. ***$p \leq 0.001$ significance between a condition and WT control by unpaired students t-test.

## CypD KO mice have numerous downregulated genes related to hepatocellular carcinoma relative to WT

To investigate the effect of CypD on the expression of genes that might be involved in NAFLD/NASH disease progression to HCC, we performed PCR arrays comparing RNA

transcripts isolated from the livers of Set 3 WT or CypD KO mice. Consistent with the results above, we found that nearly all differentially expressed genes known to be important in HCC were downregulated in the CypD KO mice (Fig 5). Notable differentially regulated genes are summarized in Fig 5. The most differentially expressed gene was Igfbp1 encoding insulin-like growth factor binding protein 1, which is expressed predominantly in the liver and is known to promote IGF signaling, a pathway deregulated in HCC [45]. The possibility that these genes are influenced by CypD should be further investigated in the future.

Set 3 WT and CypD KO mice in which pancreatic-beta cells had been disrupted by administration of STZ were nourished with western diet chow and sugar solution for thirty weeks. RNA isolated from the livers of each group were analyzed in PCR arrays for genes relevant to HCC. Green dots represent gene transcripts induced in Set 3 CypD KO mouse livers, while red dots represent decreased expression. Notably perturbed genes are summarized in the accompanying table.

## Discussion

As previous reports have shown, pan-cyclophilin inhibitor drugs can limit numerous features of late-stage NAFLD/NASH. This study underlines the importance of cyclophilins in the

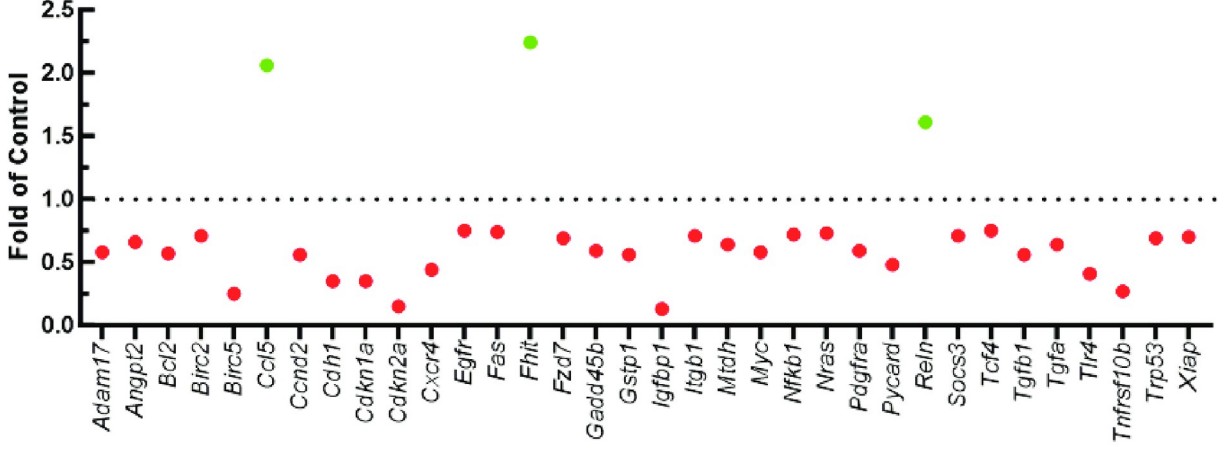

## Set 3 Mice
### PCR Array for Hepatocellular Carcinoma Genes

| Gene | Fold of Control |
|---|---|
| Fhit | 2.24 |
| Ccl5 | 2.06 |
| Reln | 1.61 |
| Pycard | 0.48 |
| Cxcr4 | 0.44 |
| Tlr4 | 0.41 |
| Cdh1 | 0.35 |
| Cdkn1a | 0.35 |
| Tnfrsf10b | 0.27 |
| Birc5 | 0.25 |
| Cdkn2a | 0.15 |
| Igfbp1 | 0.13 |

**Fig 5. CypD KO mice downregulate HCC-related genes relative to WT mice undergoing a NAFLD/NASH model.**

progression of NAFLD/NASH and highlights that CypD is of particular importance in the development of HCC. We elected to use the STZ-WD NAFLD/NASH model because it had been reported to reliably result in HCC nodules after 20 weeks. Previous experience gave us reason to extend the time frame to 30 weeks to ensure that all WT mouse livers would show evident HCC nodules. Additionally, due to constraints on the age at which we could genotype new mice to verify that they were *Ppif-/-*, we were unable to perform the more widely used STZ-WD model in which P2 neonates are injected with a small, single bolus, dose of STZ. Instead, we IP injected 4-week-old mice with a single large 200g STZ dose, coinciding with when they were switched to WD chow and sugar solution. This method has been reported to produce NAFLD/NASH disease progression similar to the neonatal method [46] and the WT and CypD KO mice tolerated the insult well despite the length of the experiment.

While we hypothesized that CypD deficiency might reduce or delay the onset of HCC, resulting in livers that had HCC but with lower scores, we were still surprised by the result that 8 of 10 CypD KO mice had no HCC whatsoever. This effect was not due to a lack of overall NASH, as STZ-WD CypD KO livers had a NAFLD Activity Score indicative of NASH and indistinguishable from WT. Liver fibrosis was similarly elevated in KO livers as was seen in WT livers. This suggests that the lack of CypD influences HCC development especially. Because of high mortality in other cyclophilin knockout lines subjected to the STZ-WD model, we could not analyze the development of NASH or HCC in those mice and thus we cannot say positively that CypD inhibition is solely responsible for decreasing tumor burden with CRV431 or NV556, which inhibit all cyclophilins. CypD ablation is sufficient to prevent HCC in this model however and given the unique subcellular localization of CypD and its role, unlike other cyclophilins, in the mPTP, it is likely that if depletion of another cyclophilin also reduced HCC burden it would be through a different mechanism.

To our knowledge, this is the first study suggesting a role specifically for CypD in promoting the development of HCC. To date, the effect of CypD on other cancer types is unclear at best and evidence exists that CypD can both promote and inhibit tumor initiation and growth, depending on the model used. For instance, CypD promotes aerobic glycolysis by recruiting one of the rate-limiting enzyme subunits, hexokinase II, to the mitochondrial outer membrane [36, 40]. Aerobic glycolysis was first discovered in HCC tumor cells, which, like other cancers, convert glucose and pyruvate into lactate even in the presence of oxygen [47]. By promoting this process CypD may play a critical role in cancer cell metabolism. CypD may also suppress cell death pathways, thereby promoting cancer cell survival, by binding and restricting the release of pro-apoptotic factors like Bcl-2 or by suppressing necrosis in response to oxidative stress [41, 48]. Conversely, CypD has also been shown to promote cancer cell death by enhancing both mPTP-mediated apoptosis [49] and mPTP-mediated necrosis pathways, in some cases through binding the tumor suppressor p53 [50]. The dual role CypD has in promoting and suppressing tumor growth [48] likely reflects its multitude of binding partners which affect mitochondrial permeability under a variety of conditions across many tissues and cancer types. Future studies on whether CypD and cyclophilin inhibiton in general are viable drug targets for the treatment or prevention of cancer should be sure to recognize that the effect of the intervention may change depending on the paradigm involved.

## Author Contributions

**Conceptualization:** Winston T. Stauffer, Daren R. Ure, Robert T. Foster, Philippe Gallay.

**Data curation:** Winston T. Stauffer.

**Formal analysis:** Daren R. Ure.

**Funding acquisition:** Philippe Gallay.

**Investigation:** Winston T. Stauffer, Michael Bobardt.

**Methodology:** Winston T. Stauffer, Philippe Gallay.

**Project administration:** Philippe Gallay.

**Resources:** Philippe Gallay.

**Visualization:** Winston T. Stauffer.

**Writing – original draft:** Winston T. Stauffer.

**Writing – review & editing:** Winston T. Stauffer, Daren R. Ure, Philippe Gallay.

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
