## [Decision Letter · Decision Letter 0]

6 Feb 2024

PONE-D-23-41197Cyclophilin D knockout significantly prevents HCC development in a streptozotocin-induced mouse model of diabetes-linked NASH

PLOS ONE

Dear Dr. Stauffer,

Thank you for submitting your manuscript to PLOS ONE. After careful consideration, we feel that it has merit but does not fully meet PLOS ONE’s publication criteria as it currently stands. Therefore, we invite you to submit a revised version of the manuscript that addresses the points raised by the reviewer #1 during the review process.

Your manuscript was found to be interesting for the field. However, several points raised by the reviewer #1 need to be addressed. Please, discuss the role of CypD in your paradigm, considering excessive cellular death in CypD KO tissue (Reviewer #2).

Please submit your revised manuscript by Mar 22 2024 11:59PM. If you will need more time than this to complete your revisions, please reply to this message or contact the journal office at plosone@plos.org. Please include the following items when submitting your revised manuscript:A rebuttal letter that responds to each point raised by the academic editor and reviewer(s). You should upload this letter as a separate file labeled 'Response to Reviewers'.A marked-up copy of your manuscript that highlights changes made to the original version. You should upload this as a separate file labeled 'Revised Manuscript with Track Changes'.An unmarked version of your revised paper without tracked changes. You should upload this as a separate file labeled 'Manuscript'.

We look forward to receiving your revised manuscript.

Kind regards,

Vadim Ten

Academic Editor

PLOS ONE

Journal Requirements:

2. Please expand the acronym “NIH/NIAID” (as indicated in your financial disclosure) so that it states the name of your funders in full.

"Daren R. Ure and Robert T. Foster are employees of Hepion Pharmaceuticals, the

developers of Reconfilstat/CRV431."

Reviewers' comments:

Reviewer's Responses to Questions

**Comments to the Author**

1. Is the manuscript technically sound, and do the data support the conclusions?

Reviewer #1: Partly

Reviewer #2: Yes

2. Has the statistical analysis been performed appropriately and rigorously? 

Reviewer #1: I Don't Know

Reviewer #2: Yes

3. Have the authors made all data underlying the findings in their manuscript fully available?

Reviewer #1: Yes

Reviewer #2: Yes

4. Is the manuscript presented in an intelligible fashion and written in standard English?

Reviewer #1: No

Reviewer #2: Yes

5. Review Comments to the Author

Reviewer #1: The format of the presented manuscript is confusing. This reviewer failed to find individual figure legends, since it seems that they are randomly inserted to the main body of the manuscript. The labelling is confusing, since all the Sets (1,2,3) are different in different figures, but not properly explained in the figure legends (assuming these were figure legends). I will be willing to read the ms after reformatting and clarification of the experimental groups assignment. Maybe it would be better to explain the Sets directly in the figure panels.

Reviewer #2: Overall, this is very interesting manuscript. The experimental plan is sounds and data presented are clear. My only concern is the author's conclusion regarding participation of the mPTP in the observed effects. According to the data CypD KO tissues (which presumable have inhibited mPTP) have even more death then WT tissues, which actually argues against the mPTP which is known to be a strong activator of cell death. This is in mind I believe that authors should discuss the possibility of physiological roles of CypD, which are independent of mPTP, which would explain the effect that they observe.

6. PLOS authors have the option to publish the peer review history of their article (what does this mean?). If published, this will include your full peer review and any attached files.

Reviewer #1: No

Reviewer #2: No

---

## [Author Response · Author response to Decision Letter 0]

13 Feb 2024

Dear PLOS One Editors and Reviewers:

We thank the editors and reviewers for taking the time to carefully read and comment on our manuscript. Below are each of the journal requirements and reviewer comments with our responses following. For clarity, our responses begin and end with a * symbol.

Journal Requirements:

*Thank you, we have made corrections as per the style requirements.*

2. Please expand the acronym “NIH/NIAID” (as indicated in your financial disclosure) so that it states the name of your funders in full.

*NIH/NIAID stands for National Institutes of Health/National Institute of Allergy and Infectious Diseases. We will wait for PLOS One to change the online submission form on our behalf. Thanks!*

"Daren R. Ure and Robert T. Foster are employees of Hepion Pharmaceuticals, the

developers of Reconfilstat/CRV431."

*Daren Ure and Robert Foster were employees of Hepion Pharmaceuticals at the time of writing of this manuscript. This does not alter our adherence to PLOS ONE policies on sharing data and materials. Additionally, the manuscript does not contain any data produced or funded by Hepion Pharmaceuticals, or any other commercial entity. and the authors thus are not reporting any competing interests, see the Competing Interests statement below.*

*Competing Interests Statement:

The authors have no conflicts to declare.*

*Because the data is not essential to the research presented here, we have elected to remove the wording mentioning the irrelevant data.*

*Thank you, we have updated our ethics statement to include our Animal Use Protocol in the Methods section.*

Reviewers' comments:

Reviewer's Responses to Questions

Comments to the Author

1. Is the manuscript technically sound, and do the data support the conclusions?

Reviewer #1: Partly

Reviewer #2: Yes

*Thank you for your comments. See below for our responses to “5. Review Comments to the Author”.*

2. Has the statistical analysis been performed appropriately and rigorously?

Reviewer #1: I Don't Know

Reviewer #2: Yes

*Thank you, see below for our responses to “5. Review Comments to the Author”.*

3. Have the authors made all data underlying the findings in their manuscript fully available?

Reviewer #1: Yes

Reviewer #2: Yes

*Thank you for your responses.*

4. Is the manuscript presented in an intelligible fashion and written in standard English?

Reviewer #1: No

Reviewer #2: Yes

*Thank you, please see below for our responses to “5. Review Comments to the Author”.*

5. Review Comments to the Author

Reviewer #1: The format of the presented manuscript is confusing. This reviewer failed to find individual figure legends, since it seems that they are randomly inserted to the main body of the manuscript. The labelling is confusing, since all the Sets (1,2,3) are different in different figures, but not properly explained in the figure legends (assuming these were figure legends). I will be willing to read the ms after reformatting and clarification of the experimental groups assignment. Maybe it would be better to explain the Sets directly in the figure panels.

*Thank you for your valuable comments. Rather than being randomly inserted in the manuscript, the figure legends are inserted after the paragraph in which the corresponding figure is first cited. While we regret any confusion, we are unable to change this formatting because it is a requirement of the PLOS One editors (as is all formatting in the manuscript). Figure citations in the text are in parentheses and read (Fig. #). Figure legends are a separate paragraph from the main text and are identifiable by bolded text that begins “Figure #.” followed by a bolded sentence corresponding to the figure’s title. The full legend then follows in regular unbolded text. I am copying the Figure 1 legend directly from the manuscript as an example:

Figure 1.

CypD KO mice have similar liver morphology to WT mice except for the appearance of HCC tumors in the STZ-WD model.

WT and CypD KO mice were separated into three sets. Set 1 mice were naïve, Set 2 received twice weekly CCl4 via intraperitoneal (IP) injection for thirty weeks, and Set 3 mice received a single STZ IP injection and were maintained on western diet (WD) for thirty weeks. Mice were sacrificed and then weighed, and the livers removed, weighed, and imaged. Set 1 livers were typically dark red with smooth surfaces. Set 2 livers were similar in size and color but had a slightly scaly surface. Many of the livers in Set 3 developed HCC tumors. Set 3 livers without tumors were larger and tan in color. Set 3 livers with tumors tended to be much smaller. Likewise, Set 3 mice tended to be larger overall, but mice with extensive HCC weighed less than mice without HCC. Representative livers for each group are shown here. ***p≤0.001 significance between a condition and WT control by unpaired students t-test.

The actual figures are submitted separately as TIF files, again, as required by the journal. We have double checked and confirmed that Sets 1, 2, and 3 refer to the same sets of mice in all figures. We added extra labeling to two of the figures to make this abundantly clear. To reiterate, Set 1 mice were nourished on ad libitum normal water and chow and served as a non-diseased baseline. Set 2 mice also received normal water and chow but was subjected to a carbon tetrachloride (CCl4) model of liver fibrosis. Set 3 mice were tested in a diabetes-linked model of NAFLD/NASH. Hopefully identifying the figure legends will aid in understanding the manuscript, as Reviewer 2 was able to do. If the PLOS One editors can provide any clarifying comments regarding how required formatting affected Reviewer 1’s understanding of the manuscript, we invite them to do so.*

Reviewer #2: Overall, this is very interesting manuscript. The experimental plan is sounds and data presented are clear. My only concern is the author's conclusion regarding participation of the mPTP in the observed effects. According to the data CypD KO tissues (which presumable have inhibited mPTP) have even more death then WT tissues, which actually argues against the mPTP which is known to be a strong activator of cell death. This is in mind I believe that authors should discuss the possibility of physiological roles of CypD, which are independent of mPTP, which would explain the effect that they observe.

*Thank you, we appreciate your kind comments. While we do make mention of CypD as a member of the mPTP complex, we do not make any claims regarding increased cell death in CypD KO tissues. Indeed, the fact that CypD global KO mice are viable at all would argue against this. To our knowledge, the literature on the role of CypD in promoting or suppressing cancer cell death is ambiguous at best. We would refer the reviewer to the last paragraph of the discussion section where we cite literature referencing CypD as both a promoter of cancer cell survival (for example by binding and sequestering BCL-2) and a promoter of cell death via the mPTP. We have added additional citations to emphasize this paradoxical relationship. While future investigation into whether there is increased HCC cell death in CypD KO mice may be warranted, it is beyond the scope of this particular manuscript.*

*Thank you again to all the PLOS One Editors and the reviewers for suggesting improvements to the overall manuscript. The article has been enhanced by your contributions. Please contact the corresponding author(s) if you have any further questions or comments.*

Sincerely,

Winston Stauffer Ph.D.

Gallay Lab

Scripps Research

La Jolla, CA

---

## [Decision Letter · Decision Letter 1]

6 Mar 2024

PONE-D-23-41197R1Cyclophilin D knockout significantly prevents HCC development in a streptozotocin-induced mouse model of diabetes-linked NASHPLOS ONE

Dear Dr. Gallay,

Thank you for submitting your manuscript to PLOS ONE. After careful consideration, we feel that it has merit but does not fully meet PLOS ONE’s publication criteria as it currently stands. Therefore, we invite you to submit a revised version of the manuscript that addresses the points raised during the review process.

We look forward to receiving your revised manuscript.

Kind regards,

Vadim Ten MD, PhD

Academic Editor

PLOS ONE

Journal Requirements:

Reviewers' comments:

Reviewer's Responses to Questions

**Comments to the Author**

1. If the authors have adequately addressed your comments raised in a previous round of review and you feel that this manuscript is now acceptable for publication, you may indicate that here to bypass the “Comments to the Author” section, enter your conflict of interest statement in the “Confidential to Editor” section, and submit your "Accept" recommendation.

Reviewer #1: All comments have been addressed

Reviewer #2: (No Response)

2. Is the manuscript technically sound, and do the data support the conclusions?

Reviewer #1: Yes

Reviewer #2: Yes

3. Has the statistical analysis been performed appropriately and rigorously? 

Reviewer #1: Yes

Reviewer #2: I Don't Know

4. Have the authors made all data underlying the findings in their manuscript fully available?

Reviewer #1: Yes

Reviewer #2: Yes

5. Is the manuscript presented in an intelligible fashion and written in standard English?

Reviewer #1: Yes

Reviewer #2: Yes

6. Review Comments to the Author

Reviewer #1: This reviewer is not a specialist in liver physiology or tumor formation. The presented results are sound and data are displayed in a proper manner.

CypD is a mitochondrial component - it would be good to present any mechanistical explanations of the observed effects on CypD in tumorigenesis It would be good to supplement these nice observational studies with some experiments aiming on elucidation of the molecular mechanism(s). Or at least separate the mitochondrial effects on MPTP CypD from other members of Pro-isomerases.

Reviewer #2: This referee's comments have been largely addressed with notable exception.

In the abstract authors state: "Both CRV431 and NV556 inhibit

several cyclophilin isoforms, among which cyclophilin D (CypD), an essential part of

the mitochondrial permeability transition pore (mPTP) complex, has not been

previously investigated in this context."

I believe this statement needs to be clarified. As authors would probably agree this study doesn't really investigate mPTP but rather reports changes associated with CypD KO with no direct experiments focused on mPTP. With this respect mention of mPTP in the abstract as "has not been previously investigated in this context" is misleading, since similar to the previous studies, this study doesn't address it either. As such, mPTP "possibility" belongs to the discussion rather than to the abstract section.

7. PLOS authors have the option to publish the peer review history of their article (what does this mean?). If published, this will include your full peer review and any attached files.

Reviewer #1: No

Reviewer #2: No

---

## [Author Response · Author response to Decision Letter 1]

8 Mar 2024

Dear PLOS One Editors and Reviewers:

We thank the editors and reviewers for taking the time to again read and comment on our manuscript. Below are each of the journal requirements and reviewer comments with our responses following in blue:

Journal Requirements:

Thank you, we have checked that the reference list is correct and does not include any retracted papers.

Reviewers' comments:

Reviewer's Responses to Questions

Comments to the Author

1. If the authors have adequately addressed your comments raised in a previous round of review and you feel that this manuscript is now acceptable for publication, you may indicate that here to bypass the “Comments to the Author” section, enter your conflict of interest statement in the “Confidential to Editor” section, and submit your "Accept" recommendation.

Reviewer #1: All comments have been addressed

Reviewer #2: (No Response)

Thank you. See below for our responses to “6. Review Comments to the Author”.

2. Is the manuscript technically sound, and do the data support the conclusions?

Reviewer #1: Yes

Reviewer #2: Yes

Thank you.

3. Has the statistical analysis been performed appropriately and rigorously?

Reviewer #1: Yes

Reviewer #2: I Don't Know

Thank you for your comments. See below for our responses to “6. Review Comments to the Author”.

4. Have the authors made all data underlying the findings in their manuscript fully available?

Reviewer #1: Yes

Reviewer #2: Yes

Thank you.

5. Is the manuscript presented in an intelligible fashion and written in standard English?

Reviewer #1: Yes

Reviewer #2: Yes

Thank you.

6. Review Comments to the Author

Reviewer #1: This reviewer is not a specialist in liver physiology or tumor formation. The presented results are sound and data are displayed in a proper manner.

CypD is a mitochondrial component - it would be good to present any mechanistical explanations of the observed effects on CypD in tumorigenesis It would be good to supplement these nice observational studies with some experiments aiming on elucidation of the molecular mechanism(s). Or at least separate the mitochondrial effects on MPTP CypD from other members of Pro-isomerases.

Thank you for your comments. As you note and as we go over in the discussion section of the manuscript, CypD is a part of the complex forming the mPTP and thus is known under varying conditions to regulate mitochondrial permeability. However, we mention CypD is part of the mPTP only to discuss obvious avenues for future investigation. We do not claim that CypD has an effect on HCC or NAFLD/NASH progression through its role in the mPTP or its localization to mitochondria at all. Figure 5 shows that CypD KO mice have numerous perturbed genes, mostly depressed, related to HCC but mostly not directly related to the mPTP. It is thus likely that CypD KO has wide ranging effects beyond just mitochondrial permeability, only some of which have been previously reported. While future experiments will certainly be warranted to explain the precise mechanism behind CypD and cyclophilin involvement in HCC formation and progression, we believe this study is important enough to report in this journal as is, in order to guide the future studies we suggest. We also agree that isolating the effects of CypD from other cyclophilin family members is important. This study was initially intended to include comparisons with other Cyp KO mice under the same disease model. As noted in the manuscript however, these arms suffered high mortality such that they could not be included in the final comparison.

Reviewer #2: This referee's comments have been largely addressed with notable exception.

In the abstract authors state: "Both CRV431 and NV556 inhibit

several cyclophilin isoforms, among which cyclophilin D (CypD), an essential part of

the mitochondrial permeability transition pore (mPTP) complex, has not been

previously investigated in this context."

I believe this statement needs to be clarified. As authors would probably agree this study doesn't really investigate mPTP but rather reports changes associated with CypD KO with no direct experiments focused on mPTP. With this respect mention of mPTP in the abstract as "has not been previously investigated in this context" is misleading, since similar to the previous studies, this study doesn't address it either. As such, mPTP "possibility" belongs to the discussion rather than to the abstract section.

Thank you for your valuable input. We agree with your concern and we have edited the abstract to remove mention of the mPTP, since as you note, it is not addressed in this study.

7. PLOS authors have the option to publish the peer review history of their article (what does this mean?). If published, this will include your full peer review and any attached files.

Do you want your identity to be public for this peer review? For information about this choice, including consent withdrawal, please see our Privacy Policy.

Reviewer #1: No

Reviewer #2: No

Thank you again to all the PLOS One Editors and the reviewers for your comments and improvements to the overall manuscript. The article has been further enriched by your contributions. Please contact the corresponding author(s) if you have any further questions or comments.

Sincerely,

Philippe Gallay

Professor

Department of Immunology & Microbiology 

The Scripps Research Institute

10550 North Torrey Pines Road

La Jolla, California 92037

Tel. (858) 784 8180

Fax (858) 784 8831

http://www.scripps.edu/gallay/

---

## [Decision Letter · Decision Letter 2]

20 Mar 2024

Cyclophilin D knockout significantly prevents HCC development in a streptozotocin-induced mouse model of diabetes-linked NASH

PONE-D-23-41197R2

Dear Dr. Philippe Gallay

We’re pleased to inform you that your manuscript has been judged scientifically suitable for publication and will be formally accepted for publication once it meets all technical requirements.

Kind regards,

Vadim Ten MD, PhD

Academic Editor

PLOS ONE

Additional Editor Comments (optional):

Reviewers' comments:

Reviewer's Responses to Questions

**Comments to the Author**

1. If the authors have adequately addressed your comments raised in a previous round of review and you feel that this manuscript is now acceptable for publication, you may indicate that here to bypass the “Comments to the Author” section, enter your conflict of interest statement in the “Confidential to Editor” section, and submit your "Accept" recommendation.

Reviewer #1: All comments have been addressed

Reviewer #2: All comments have been addressed

2. Is the manuscript technically sound, and do the data support the conclusions?

Reviewer #1: Yes

Reviewer #2: Yes

3. Has the statistical analysis been performed appropriately and rigorously? 

Reviewer #1: Yes

Reviewer #2: Yes

4. Have the authors made all data underlying the findings in their manuscript fully available?

Reviewer #1: Yes

Reviewer #2: Yes

5. Is the manuscript presented in an intelligible fashion and written in standard English?

Reviewer #1: Yes

Reviewer #2: Yes

6. Review Comments to the Author

Reviewer #1: All issues were addressed in this revision. The paper can be accepted for publication in the presented format.

Reviewer #2: all good. concerns have been addressed. I have no further comments. I thank authors for their excellent work.

7. PLOS authors have the option to publish the peer review history of their article (what does this mean?). If published, this will include your full peer review and any attached files.

Reviewer #1: No

Reviewer #2: No

---

## [Editor Report · Acceptance letter]

26 Mar 2024

PONE-D-23-41197R2 

PLOS ONE

Dear Dr. Gallay, 

I'm pleased to inform you that your manuscript has been deemed suitable for publication in PLOS ONE. Congratulations! Your manuscript is now being handed over to our production team.

Kind regards, 

on behalf of

Professor Vadim Ten 

Academic Editor

PLOS ONE